# Gradient descent in materia through homodyne gradient extraction

Marcus N. Boon [1,2,3,6], Lorenzo Cassola[1,4,6], Hans-Christian Ruiz Euler[1], Tao Chen [1], Bram van de Ven[1], Unai Alegre Ibarra[1], Peter A. Bobbert [1,5] & Wilfred G. van der Wiel [1,4] ✉

Deep learning, a multilayered neural-network approach inspired by the brain, has revolutionized machine learning. Its success relies on backpropagation, which computes gradients of a loss function for use in gradient descent. However, digital implementations are energy hungry, with power demands limiting many applications. This has motivated specialized hardware, from neuromorphic CMOS and photonic tensor cores to unconventional material-based systems. Learning in such systems, for example via artificial evolution, equilibrium propagation, or surrogate modelling, is typically complicated and slow. Here, we demonstrate a simple gradient-extraction method based on homodyne detection, enabling gradient descent directly in physical systems without the need for an analytical description. By perturbing parameters with sinusoidal waveforms at distinct frequencies, we robustly obtain gradient information in a scalable manner. We illustrate the method in reconfigurable nonlinear-processing units and argue for broad applicability. Homodyne gradient extraction can in principle be fully implemented in materia, facilitating autonomously learning material systems.

Complex optimization problems, such as those occurring in signal processing and machine learning, are often hard to solve analytically and become increasingly time and energy consuming as the dimensionality of the parameter space increases. In deep learning[1], the input-output relation of a deep neural network (DNN) model is given by a differentiable, multi-variable function, where gradient information can be readily obtained from the partial derivatives of the output with respect to the DNN parameters (weights and biases). Backpropagation[2] is used to efficiently compute the gradient of a pre-defined loss function with respect to the DNN parameters, which allows for training complex DNNs with thousands to billions of parameters. By means of iterative, gradient-based optimization methods, such as stochastic gradient descent[3], DNNs are trained to show unprecedented performance, but often at a high, financial and environmental, cost[4]. Given

the rising costs of digitally implemented neural networks, there has been an increasing interest in training and deploying physical neural networks (PNNs)[5–9], a class of materials systems that leverage the physical properties of "intelligent matter"[10] to perform information processing[11]. This, in turn, has given rise to a plethora of newly developed methods for training PNNs, which can be separated into two main categories: model-based optimization (MBO) methods, where a model of (a part of) the physical system is used to perform the parameter optimization, and model-free optimization (MFO) methods, where no complete or only a partial analytical description of the system is needed.

In MBO methods, including in-silico training (e.g., ref. 12), physics-aware learning[13], and feedback alignment[14] and its derivates[15,16], a software model or a generic system approximation is used to extract

[1]NanoElectronics Group, MESA+ Institute for Nanotechnology and BRAINS Center for Brain-Inspired Computing, University of Twente, Enschede, The Netherlands. [2]Exzellenzcluster Science of Intelligence, Technische Universität Berlin, Berlin, Germany. [3]Department for Electrical Engineering and Computer Science, Modeling of Cognitive Processes, Technische Universität Berlin, Berlin, Germany. [4]Institute of Physics, University of Münster, Münster, Germany. [5]Molecular Materials and Nanosystems & Eindhoven Institute for Renewable Energy Systems, Department of Applied Physics, Eindhoven University of Technology, Eindhoven, The Netherlands. [6]These authors contributed equally: Marcus N. Boon, Lorenzo Cassola. ✉e-mail: W.G.vanderWiel@utwente.nl

gradients during training. Hence, an external (digital) computer is always required for training. Although this allows for accurate training (even if accuracy degradation cannot always be avoided[5]), it brings along severe limitations. The use of MBO methods can be challenging in the absence of an analytical model, as it requires developing a numerical model tailored to each specific system. Such numerical models often demand substantial computational resources and frequently struggle to capture the physical system's nonidealities accurately[17,18]. Furthermore, the applicability of feedback alignment-based techniques is limited in certain physical systems, particularly when separating the linear and nonlinear components of the input-output relationship is not feasible[5]. On-the-fly, in-hardware optimization, essential for applications requiring fast and low-power reconfigurability, is inherently impossible in MBO methods. All this makes MBO methods cumbersome and case-specific, preventing autonomous learning, i.e., learning without the assistance of an external system or operator. This is particularly problematic for edge computing[19–21] as edge devices need to perform tasks and make decisions independently, without continuous human intervention or reliance on centralized cloud resources[22].

MFO methods, which do not rely on a model, can on their turn, be divided into two subclasses: on the one hand gradient-free optimization (GFO) methods, and on the other hand approaches that—although model-free—are gradient-based. GFO is an umbrella term for a class of algorithms that iteratively generate better solutions to specific optimization problems through population-based sampling[5]. This class includes evolutionary strategies, such as genetic algorithms[20,23], and swarm optimization[24,25]. Despite their broad use, GFO methods are often unsuitable for large-scale problems due to high computational demands, and do not allow for convergence-rate analysis[26]. In the remaining category of model-free, gradient-based methods, notable methods include physical local learning (PLL)[27], equilibrium propagation (EP)[28], and zeroth-order optimization (ZO) methods, which optimize a system solely based on its inputs and outputs, without requiring explicit gradient information[26]. Also, these methods have their limitations. PLL was introduced as a training method for deep PNNs composed of multiple physical layers, each one with its own local loss function to be optimized. However, updating the parameters of every physical layer requires accurate estimation of the local gradients, which can be complicated to evaluate in disordered systems and emerging device platforms that have recently gained popularity for physical optimization[29,30]. EP is not suitable for every physical system, primarily because it requires stable equilibrium states to effectively minimize an energy or Lyapunov function. Additionally, the method's reliance on having two identical copies of the system for gradient estimation can be impractical in systems where precise replication is difficult or impossible[5]. Finally, ZO methods are generally slow, since the number of gradient updates scales linearly with the number of weights and biases[5]. In addition, ZO optimization methods tend to be very sensitive to noise, particularly at low frequencies, which makes their application to physical systems, often exhibiting $1/f$-like noise, challenging. The most prominent examples of ZO gradient approximation methods are finite difference (FD) and simultaneous perturbation stochastic approximation (SPSA)[31]. FD perturbs one parameter at a time and estimates each individual gradient component by computing the difference in function values, requiring a separate function evaluation per parameter. In contrast, SPSA perturbs all parameters simultaneously along a random direction and uses the resulting approximate gradient for optimization. SPSA thus reduces computational overhead by requiring, in principle, only two function evaluations, at the cost of greatly reduced gradient accuracy. Overcoming the drawbacks of ZO methods and finding a balance between computational cost and accuracy would allow us to exploit their key advantage: the need for only forward evaluations.

Here, we demonstrate an efficient gradient extraction method for ZO optimization of complex (viz., multi-parameter and nonlinear) physical systems in general, and electronic systems in particular. We refer to our method as homodyne gradient extraction (HGE), as we use homodyne (or lock-in) detection[32] to frequency-selectively determine the response to sinusoidal perturbations on all input parameters of the system at the same time, allowing us to extract the gradients of the system response to the respective parameters. By exploiting the advantages of lock-in detection, gradients can be accurately extracted in a noisy system, even with a dominant $1/f$-like contribution, and for multiple parameters in parallel without loss in accuracy. Moreover, HGE can in principle be fully realized in electronics, i.e., without digital signal processing, enabling on-the-fly optimization without external support.

We found HGE to perform quick and robust gradient evaluations, allowing us to train nanoelectronic devices with a significant gain in terms of both speed and accuracy compared to the FD method. For the devices considered here, HGE presents a gain in terms of time proportional to the number of parameters to be optimized, while the variance of the obtained gradient is shown to be about two orders of magnitude smaller, thanks to the particular suitability of our approach for noisy systems, specifically for systems characterized by $1/f$-like noise.

## Results

### General concept of HGE

The general HGE concept is schematically presented in Fig. 1a for an arbitrary multi-parameter system represented by the function $h(\mathbf{z}, \mathbf{w})$ with $N$ inputs represented by the vector $\mathbf{z}$ ($z_1, ..., z_N$) and $M$ optimizable parameters represented by the vector $\mathbf{w}$ ($w_1, ..., w_M$). We assume that the system lacks an analytical description, so that computing the derivatives with respect to the optimizable parameters is not possible. Instead, only direct system evaluations are possible, which are assumed to be inherently noisy (e.g., due to thermal or $1/f$-like noise). Now, to optimize the system, a loss function $E(\mathbf{w})$ is defined, quantifying the difference between the actual system output $h(\mathbf{z}, \mathbf{w})$ and the desired output. Since $E(\mathbf{w})$ inherently depends on $h(\mathbf{z}, \mathbf{w})$, which in turn depends on the parameters $\mathbf{w}$, optimizing $E(\mathbf{w})$ requires the computation of its gradient with respect to $\mathbf{w}$. According to the chain rule, this gradient can be expressed as a product of two terms: the derivative of $E(\mathbf{w})$ with respect to the output, $\partial E/\partial h$, and the derivatives of the output with respect to each parameter $w_m$, collectively forming the gradient $\partial h/\partial w_m$. The essence of HGE is that $\partial h/\partial w_m$ does not need to be calculated using an analytical description of the system, but instead can be straightforwardly approximated using homodyne (or "lock-in") detection. Perturbations $\delta w_m(t) = \alpha_m \sin(2\pi f_m t + \varphi_m)$ are added to all parameters $w_m$ in parallel, where each perturbation has a distinct frequency $f_m$ and phase $\varphi_m$, and optionally a distinct amplitude $\alpha_m$, and the corresponding changes in $h(\mathbf{z}, \mathbf{w} + \boldsymbol{\delta w})$ are measured (Fig. 1a). By using distinct frequencies and/or phases, we can extract each signal independently through the lock-in detection principle, as long as the frequencies $f_m$ are sufficiently separated (see below).

The output of the perturbed system, $h(\mathbf{z}, \mathbf{w} + \boldsymbol{\delta w})$, is mixed with sinusoidal reference signals that are in-phase and 90° phase-shifted (quadrature) relative to the parameter perturbations. The reference signals use the same frequencies $f_1, ..., f_M$ and phases $\varphi_1, ..., \varphi_M$ as the perturbations. This mixing shifts the frequency spectrum of the resulting signals by their respective reference frequencies in both the positive and negative directions (Fig. 1b, rightmost panel). For each parameter $w_m$, the mixed signal at the corresponding reference frequency $f_m$ produces a 0 Hz (DC) component. This DC component is directly proportional to the changes in $h(\mathbf{z}, \mathbf{w} + \boldsymbol{\delta w})$ caused by the sinusoidal perturbation of $w_m$ and is used for calculating the derivative with respect to parameter $w_m$. To extract it from the total signal, a low-

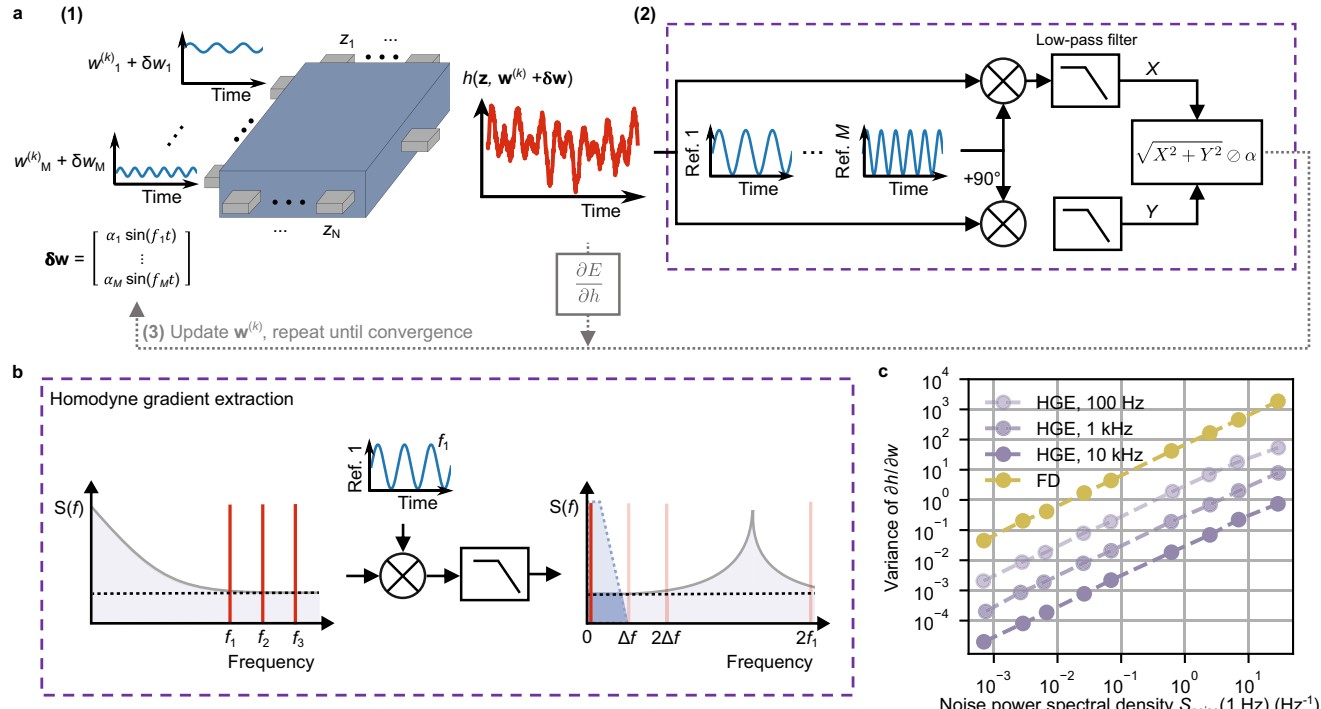

**Fig. 1 | Principle of homodyne gradient extraction (HGE). a (1)** Small sinusoidal signals (with frequencies $[f_1, ..., f_M]$, amplitudes $[\alpha_1, ..., \alpha_M]$, and phases $[\varphi_1, ..., \varphi_M]$) are added to the parameters $\mathbf{w}$. The output $h(\mathbf{z}, \mathbf{w} + \delta\mathbf{w})$ (red curve) is affected by each distinct sinusoidal perturbation $\delta w$. **(2)** The in-phase and quadrature components ($X$ and $Y$) of the modulation of $h$ for each frequency $f_m$ are recovered by using the principle of homodyne (lock-in) detection. These magnitudes are divided by the amplitudes of the corresponding perturbations (vector $\boldsymbol{\alpha}$), extracting the derivatives of the output with respect to each parameter, $\left[\partial h/\partial w_1^{(k)}, \cdots, \partial h/\partial w_M^{(k)}\right]$, in iteration step $k$. **(3)** The approximated gradient in step $k$ along with the derivative of the loss with respect to the output current, $\partial E/\partial h$, is used to update the parameters during the gradient descent. **b** Schematic representation of HGE in the frequency domain to extract the gradient from the spectral power density $S(f)$, with the shading indicating the noise spectrum. Distinct frequencies $f_1, f_2, f_3, ...$ with separation $\Delta f$ are used to perturb multiple parameters in parallel. After mixing the output signal with a reference signal (here shown for $f_1$), the derivative information is extracted from the total signal by removing undesired components at nonzero frequencies (i.e., noise and interference of other input perturbations) using a low-pass filter (blue shaded trapezoid). The horizontal dotted lines indicate the background white noise. **c** Effect of the magnitude of $1/f$ noise (expressed in terms of unfiltered noise at 1 Hz, $S_{noise}$ (1 Hz) on the ability to extract the derivative using FD and HGE in simulation for the function $h = 10w$ (error bars based on 1000 repetitions, $\alpha = 0.1$, $T = 1$ s, $w = 1 + \delta w$). The variance of the HGE derivative is shown for various perturbation frequencies.

pass filter is applied, evaluating the in-phase ($X_m$) and quadrature ($Y_m$) components (as shown in Fig. 1a, with the extraction process illustrated in Fig. 1b). The cut-off frequency of the low-pass filter, $f_c$, is inversely proportional to the measurement time $T$, i.e., the time needed to perform the gradient extraction ($f_c = 1/2\pi\tau$, where $\tau$ is the filter's time constant, $\tau \propto T$). Thus, the accuracy of the extraction scales with measurement time. To approximate the derivative of $h(\mathbf{z}, \mathbf{w})$ with respect to parameter $w_m$, $\partial h/\partial w_m$, the component $X_m$, and, in case of the amplitude measurement, $R_m = \sqrt{X_m^2 + Y_m^2}$, is divided by the perturbation amplitude $\alpha_m$. Next, the derivative of the loss function with respect to the output, $\partial E/\partial h$, is computed and multiplied by the estimated $\partial h/\partial w_m$ to obtain $\partial E/\partial w_m$ for each parameter. As the loss function is user-defined, its derivative with respect to the output is known and can be computed analytically. By using an optimization scheme, the parameters for iteration $k$ in the gradient descent process $\mathbf{w}^{(k)}$ are updated to $\mathbf{w}^{(k+1)}$.

Similar to FD, HGE utilizes perturbations to approximate the gradient, which is then used for optimization. However, HGE's advantage over FD in using sinusoidal, rather than step-based perturbations, is twofold. First, we can choose at what frequency we extract the gradient. If the noise spectral density is not flat, but for example has a $1/f$-like dependence[33], HGE allows to obtain the gradient at finite frequencies where the noise is much lower than close to 0 Hz. Figure 1b illustrates this principle: the output signal, containing perturbations at

frequencies $f_1, f_2$, and $f_3$ but also noise, is mixed with a reference signal of frequency $f_1$. The resulting signal has all sum and difference frequencies with respect to $f_1$, effectively shifting the output signal at $f_1$ to 0 Hz and $2f_1$, and shifting the noisiest part of the spectrum, originally located at lower frequencies, further away from 0 Hz. A low-pass filter is then used to extract the signal at 0 Hz, removing noise as well as the component at $2f_1$ and other sum and difference frequencies. FD, in contrast, necessarily operates around 0 Hz, leading to more noise and a less accurate gradient estimation. In the worst case, the noise power is as high or even higher than the power of the perturbated output signal, preventing reliable gradient estimation. By contrast, HGE can still accurately estimate the gradient, as illustrated in Fig. 1c, where we plot the simulated variance in $\partial h/\partial w$ as a function of the $1/f$-like noise power $S_{noise}$ for different perturbation frequencies. At a given $S_{noise}$ (evaluated at 1 Hz), for all frequencies, the HGE derivatives are orders of magnitude more accurate than the FD result. Furthermore, each order of magnitude increase in perturbation frequency reduces the variance by a factor of 10. Increasing the signal-to-noise-ratio can in principle also be achieved by increasing the perturbation amplitude. However, for nonlinear systems, this can lead to a bias in the gradient estimation, since the linearity assumption in the first-order Taylor expansion (see "Methods") is violated for large perturbations (example shown in Supplementary Note 1). This holds for both FD and HGE. Therefore, increasing the perturbation amplitude is often not a viable option to increase the derivative accuracy.

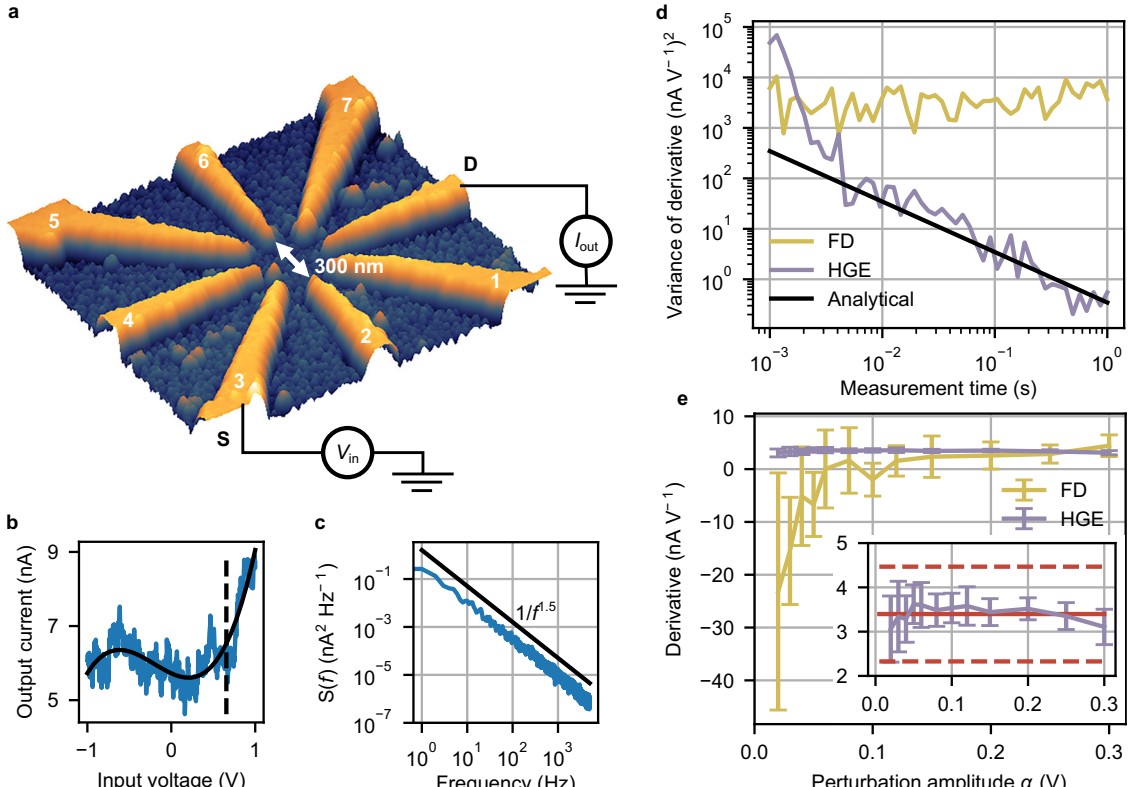

**Fig. 2 | Sequential extraction of derivatives. a** Atomic force microscope image of a reconfigurable nonlinear processing unit (RNPU). The orange regions are metallic source (S), drain (D), and control electrodes, the grey region is doped Si. **b** Single current-voltage characteristic at 77 K for the S and D electrodes in (**a**). Fixed randomly chosen voltages in the range [−1 V, 1 V] are applied to the other electrodes. The solid black curve is a 3rd-order polynomial fit to the data and the dashed line indicates the voltage of derivative extraction. **c** Power spectral density $S(f)$ of the current measured at the voltage indicated by the dashed line in (**b**). The solid black line indicates a $1/f^{1.5}$ dependence. **d** Variance of the HGE and FD derivatives (evaluated at dashed line in (**b**)) as a function of measurement time $T$, for a perturbation amplitude $\alpha = 8$ mV. The variance of the HGE derivative is compared with an analytical expression of the expected HGE error (see "Methods"). **e** Derivatives (evaluated at dashed line in (**b**)) and their standard deviations (error bars) as a function of perturbation amplitude $\alpha$ (measurement time per data point fixed at 0.1 s). Inset: zoom in around the average derivative of 3rd-order polynomial fits of 50 I-V measurements, evaluated at the dashed line in (**b**). The red solid and dashed lines represent the mean and the standard deviation of the analytical derivative of the polynomial fit, respectively.

Next to greater accuracy in the presence of $1/f$-like noise, the second advantage of HGE is parallelizability. We are free to choose the frequency and/or phase for the perturbations within a lower bound defined by the error introduced by the $2f$-component and an upper bound defined by a fraction of the maximum sampling frequency. Hence, we can encode the derivative information for the separate components of the vector $\boldsymbol{w}$ using distinct frequencies (and phases, provided that the response of the considered system has negligible time delay). Thereby, we effectively parallelize the gradient extraction procedure, as illustrated by the signals at frequencies $f_2$ and $f_3$ in Fig. 1b. This comes with additional constraints to ensure that the parallel frequencies do not introduce a bias in the gradient extraction of the other components, as discussed in detail below. To validate our approach in physical devices, we first compare sequential (i.e., single-frequency) HGE with FD and assess its accuracy gain with increased measurement time, benchmarking it against an analytical expression. Next, we experimentally and analytically demonstrate parallel (i.e., multi-frequency) HGE in real devices, identifying key factors that limit its scalability. Finally, we apply parallelized HGE gradient descent in materia to benchmark tasks, demonstrating its efficiency and effectiveness.

## Comparison of sequential HGE with FD in a physical system

In this section, we determine the measurement time required to achieve sufficient accuracy and demonstrate HGE's robustness to noise on a real device by analyzing how the variance of the estimated derivative depends on several factors. For now, we apply HGE in a sequential fashion, i.e., we determine the derivative with respect to one control parameter at a time. We discuss parallel HGE in the next section. We experimentally demonstrate this sequential HGE in a nonlinear, nanoelectronic multi-terminal device, referred to as a reconfigurable nonlinear processing unit (RNPU, Fig. 2a)[12,20,34]. The device consists of an electrically tuneable disordered network of boron dopants in silicon (Si:B) with eight terminals, seven of which act as either voltage inputs or controls (i.e., optimizable parameters), and one as current output. The output response of the device with respect to any of the inputs is in general nonlinear at 77 K (Fig. 2b, solid line: fit to the data) and exhibits $1/f$-like noise (Fig. 2c, note that the noise plateau is not reached due to the limited sampling frequency of the experimental setup). Previously, we demonstrated that the voltage controls can be trained to enable a RNPU to solve linearly inseparable classification problems or perform feature extraction[35]. Training was achieved using either an evolutionary approach[19,20,36] or gradient descent on a DNN surrogate model of the physical device[12].

In Fig. 2d we compare the estimated derivative of the RNPU output current with respect to one of its control voltages using HGE and FD for increasing measurement time $T$. The cut-off frequency of the low-pass filter (see "Methods") used for both HGE and FD is chosen such that the filter is 98% settled before taking the measurement, which for HGE is at time $t = T$, and for FD, due to requiring 2 samples, at time $t = T/2$. In addition to DC voltages applied at each input terminal,

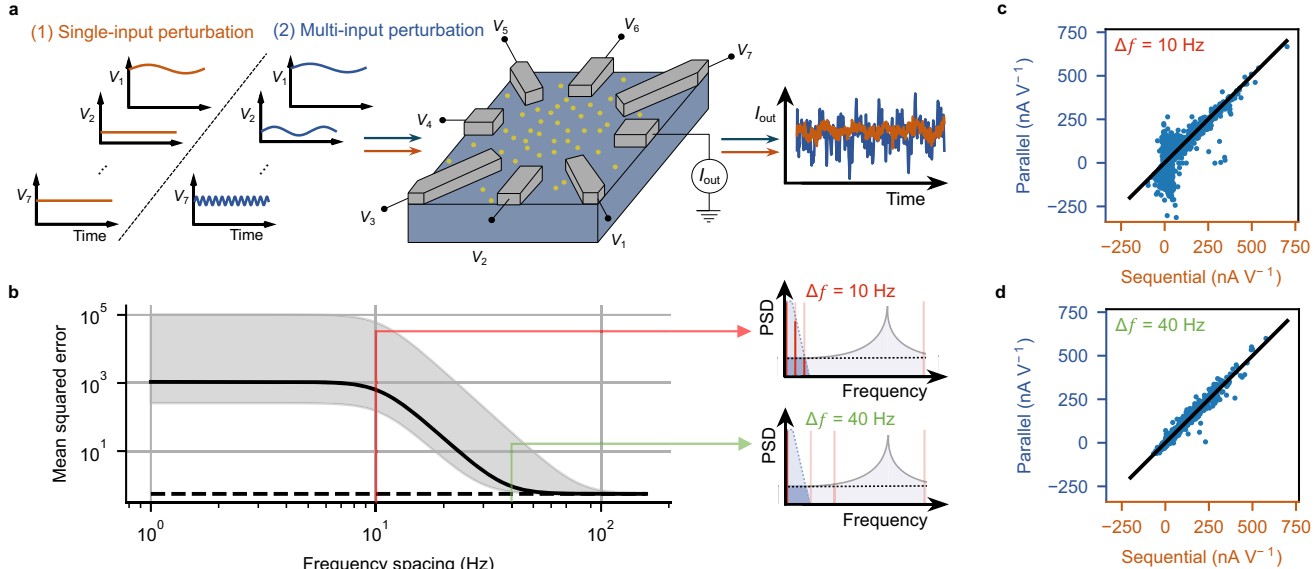

**Fig. 3 | Parallelization of HGE in a RNPU. a** Schematic representation of the RNPU (yellow dots represent boron atoms) and the two different ways to extract the gradient of the output current with respect to the input voltages using HGE: single-input and multi-input. The rightmost panel shows typical experimental output currents for both cases. **b** Analytical description of expected mean squared error *vs.* frequency spacing $\Delta f$. Horizontal dashed line: mean squared error for the sequentially extracted derivative. Solid black line: mean squared error for the derivative extracted in parallel, for a channel with an output signal strength that is 2.05 times weaker than its neighbouring frequencies. Upper limit of grey area: the same, but for a channel with an output signal strength that is 20 times weaker than

the others. Lower limit of grey area: the same, but for a channel with an output signal strength that is as strong as all the others. The magnitude of the noise is fixed to a value of $10^{-5}$ nA$^2$ Hz$^{-1}$ at 1 kHz (see Fig. 2c). Side panels: Schematic power spectral density (PSD) graphs of a parallel mixed HGE signal, demonstrating how the frequency spacing $\Delta f$ affects the final filtered signal. **c**, **d** Comparison between the single- and multi-input HGE derivatives for 1000 sets of random input voltages applied to the RNPU with a frequency spacing of either 10 Hz (**c**), or 40 Hz (**d**). In both cases the black line indicates $y = x$. The measurement time per data point is 0.1 s both in the calculations and the measurements.

we additionally perturb one input (dashed line in Fig. 2b) with either a sinusoidal function for HGE (amplitude $\alpha = 8$ mV, frequency 1 kHz) or a step function for FD with step sizes $\pm\,\alpha$ (using central FDs). We observe that the variance of the estimated derivative (over 10 repetitions) using FD remains roughly constant as a function of the measurement time $T$, whereas for HGE the variance decreases with $1/T$, i.e., it decreases with decreasing cut-off frequency of the low-pass filter. This result confirms what we stated in the previous section regarding the superior accuracy of HGE compared to FD. In addition, we developed a simple analytical model to predict how the HGE's variance depends on the measure-ment time. This model (black line in Fig. 2d), which is in good agree-ment with the experimental data, was obtained considering the impact of several factors (such as noise, the $2f$ component, and the low-pass filter) on the variance (see "Methods").

We now investigate how the perturbation amplitude affects the accuracy of the extracted derivative for both HGE and FD (Fig. 2e). The derivative is additionally estimated by fitting a third-order polynomial to the *I-V* curve (black curve in Fig. 2b) and determining the derivative of this polynomial, to provide a reference value in a comparison with HGE and FD. Note that this calculated derivative is not intended to represent a ground-truth value. Instead, it serves as an additional validation, as it is derived from a polynomial fit based on noisy data. To provide a better reference to what the "true" value of the derivative should be, we measured the *I-V* curve shown in Fig. 2b 50 times and calculated the mean and standard deviation of the derivative from the polynomial fits to each curve (red solid line and dashed lines in inset of Fig. 2e, respectively). We observe that, while the variance for HGE and FD (calculated for 10 repetitions) decreases with increasing perturba-tion amplitude, the HGE variance is drastically lower than that of the FD derivative. Furthermore, the average value of the HGE derivative matches the derivative derived from the polynomial fits much better. HGE is thus remarkably robust and outperforms FD under noisy con-ditions. We furthermore investigate how the perturbation amplitude

influences the accuracy of the extracted derivative for both HGE and FD (Fig. 2e).

## Parallelization of HGE

In the previous section, we demonstrated HGE in a RNPU for a single parameter at a time, i.e., sequential HGE. However, as shown in Fig. 1, HGE allows for simultaneous extraction of multiple derivatives at a time, thereby evaluating the full gradient. In this section we will focus on the scalability and demonstrate how HGE can be parallelized in a RNPU without loss of accuracy. In Fig. 3a we compare sequential (single-input, orange curves) to parallel (multi-input, blue curves) perturbations. For a single-input perturbation, multiplying the output by the same frequency and applying a low-pass filter allows isolating the DC component, as there are no other perturbation frequencies near zero frequency. For multi-input HGE, however, the spacing between the perturbation frequencies needs to be chosen such that the DC component can still be isolated from other perturbation fre-quencies and harmonics like the $2f$ component (Fig. 3b).

To determine the optimal spacing between the frequencies, an analytical expression of the mean squared error for the derivative extraction is derived, based on the model also used in Fig. 2d, that determines the additionally expected error due to parallelization (see "Methods"). This error depends on various factors, such as the mag-nitude of the noise, the output response to the perturbations, and the spacing $\Delta f$ between the perturbation frequencies. To set a lower boundary on the accuracy of the extracted derivative, in Fig. 3b the predicted value of the error is plotted versus $\Delta f$ for the electrode with the weakest output signal strength (in RNPUs, depending on the geometry of the device and on the set of voltages applied to it, the signals applied at different electrodes have different influences on the output signal). Some of the factors contributing to the error are device-specific and can be measured experimentally. The noise magnitude, for example, is in the calculation for Fig. 3b fixed to a value of

$10^{-5}$ nA$^2$ Hz$^{-1}$ (at 1 kHz), which is directly determined from measurements (see Fig. 2c) and is, for the sake of simplicity, assumed to be the same across the parameter space. Other factors, like the effect on the extracted derivative of the perturbations applied to the other electrodes, depend on the operational point in the parameter space and on the amplitudes chosen for the perturbations. These effects vary sensitively from case to case, and, for this reason, the calculation of the expected error was done for different ratios between the output signal strengths of all the other electrodes and that of the considered electrode. The solid black curve in Fig. 3b shows the error for the median ratio of 2.05, obtained from 1000 HGE measurements on the RNPU (see Supplementary Note 2). The upper boundary of the grey region shows the error for a large but still realistic ratio of 20 (for less than 1% of the different measured sets of voltages the ratio is larger than 20), while for the lower boundary the ratio is 1. The expected error in the parallel derivative extraction is compared to the sequential case (using the same values for noise, measurement time, etc.), indicated by the black dashed line in Fig. 3b. There are two different regimes in terms of frequency spacing $\Delta f$: in the first regime the parallel error is higher than the sequential error, and in the second regime the errors are of similar magnitude. In the first regime, the frequencies of the parallel perturbations are too closely spaced, making it impossible for the low-pass filter to remove these additional parallel signals (see Fig. 3b, upper right panel). As a result, the extracted derivatives are biased by the other perturbations. While one could increase the measurement time and hence reduce the filter's cut-off frequency to sufficiently filter the parallel frequencies, the goal of the present demonstration is to determine the spacing of the frequencies without changing the setup compared to the sequential approach. Only then can we have a fair comparison between the two and demonstrate at which frequency spacing the parallelization essentially comes "for free", i.e., without loss of accuracy. For the specific case of $\Delta f = 10$ Hz, we show from measurements on the device that the sequentially extracted derivatives do not match the derivatives extracted in parallel (Fig. 3c, the black line represents a line with unity slope). For a frequency spacing of about 40 Hz and larger, in contrast, the analytical expression in Fig. 3b predicts that the error due to parallelization is no longer dominant. The parallel extracted derivatives are then as accurate as the sequential derivatives, leading to an unbiased gradient estimation. The other perturbations are sufficiently filtered by the low-pass filter, which is schematically shown in the lower right panel of Fig. 3b. This is experimentally verified in the device, as shown in Fig. 3d.

Instead of requiring a measurement time of 0.1 s per optimizable parameter, as is done in the sequential case, all seven parameters are perturbed within a single 0.1 s measurement, effectively speeding up the gradient extraction process by a factor of 7. Note that for this particular device, this is the maximally achievable speedup since the speedup is limited by the number of electrodes. In principle, given $\Delta f = 40$ Hz, we could fit many more parallel frequencies in our frequency band, which has a lower bound defined by the measurement time of one measurement (see Fig. 2d), and an upper bound defined by the maximum achievable perturbation frequency (which is 1 kHz for our measurement setup).

**Performing benchmark tasks by gradient descent in materia**
In this section, we apply HGE to train our devices for benchmark tasks. Firstly, we focus on obtaining Boolean logic gates, since these have already been used to test the functionality of RNPUs[12] and can therefore be helpful in comparing HGE with other training procedures. Moreover, as explained in ref. 20, the X(N)OR gate is a valuable benchmark for evaluating the ability of RNPUs to perform nonlinear classification. The benchmark results, shown in Fig. 4a, are obtained using electrodes 1 and 6 (Fig. 2a) as inputs, with −0.7 V and 0.7 V representing logic labels 0 and 1, respectively. This setup enables the composition of the four logic input combinations (00, 01, 10, 11), which are applied to the device. The resulting output currents are recorded and used to evaluate the loss function. The loss function is composed of a combination of a correlation and a sigmoid function, promoting both the desired output shape and the separation between the two class labels (see "Methods"). The loss function is then minimized by means of the gradients obtained in parallel using HGE (see "Methods" for the update scheme). All the logic gates (all 16 possible truth tables) were found successfully, with a threshold chosen to lie halfway in between the average values of the current for the 1 and the 0 states, clearly separating the "high" and "low" labels. The realization of the major logic gates is shown in Fig. 4a. All the logic gates were found successfully, with a threshold chosen to lie halfway in between the average values of the current for the 1 and the 0 states, clearly separating the "high" and "low" labels. The realization of the major logic gates is shown in Fig. 4a.

To further illustrate the efficiency of our training method, we apply HGE to an additional designed task, which is a sphere classification, a 3D version of the ring classification task we presented in ref. 12. This task consists of classifying two classes of data points in a three-dimensional feature space. The first class, labelled "0", lies in an outer spherical shell, while the second class, labelled "1", consists of data points in an inner sphere. The sphere classification task requires three electrodes as inputs, leaving only four electrodes available for control, making this the most complex classification task a RNPU has been trained for thus far. Electrodes 2, 4, and 6 (see Fig. 2) were chosen as voltage inputs, with values ranging between −0.7 and 0.7 V. A similar training procedure is followed as for the Boolean logic. In Fig. 4b the results are presented in a 3D scatter plot, where the colour quantifies the output current for each of the 1000 points of the test dataset. The training dataset consisted of 300 points. In Fig. 4c the output currents for the test dataset are shown, while Fig. 4d reports the learning curve (the learning was stopped when the loss function reached the threshold of value 0.2), which decreases monotonically and shows typical behaviour for gradient descent. The separation threshold between the "0" and "1" classes was chosen to maximize the classification accuracy (94%). We furthermore note that our HGE results are more robust to those obtained by a genetic algorithm (GA) and a surrogate model. Where the GA is not able to solve the sphere classification at all, the training of a surrogate model takes far longer for a similar or even worse accuracy (see Supplementary Note 3). Lastly, in addition to comparing HGE directly on-chip, we furthermore compared the gradient accuracy and convergence properties for a benchmark task in simulation, where we tested HGE against multiple variants of SPSA. Again, we find that HGE more accurately estimates gradients across a large range of noise conditions and input dimensions, and that HGE largely outperforms SPSA on the benchmark task, requiring fewer overall samples to converge (see Supplementary Note 4).

## Discussion
We have proposed a generally applicable HGE method to obtain the gradient of the output response with respect to the input parameters of a physical system. The method was demonstrated in materia for a nanoelectronic device consisting of an electrically tuneable network of boron dopants in silicon (RNPU). Our approach can also be applied to layered networks (see Supplementary Note 5), such as PNNs, and other physical devices and systems, like quantum dot devices[37,38], integrated optical systems[39,40], and metasurfaces[41], as long as one can establish the output response with respect to perturbations in the input signals. A key advantage of HGE is the possibility of parallelization. We demonstrated this for a RNPU with 7 inputs and 1 output, but our approach is in principle applicable to any number of inputs and outputs, as long as the perturbation frequencies can be well separated.

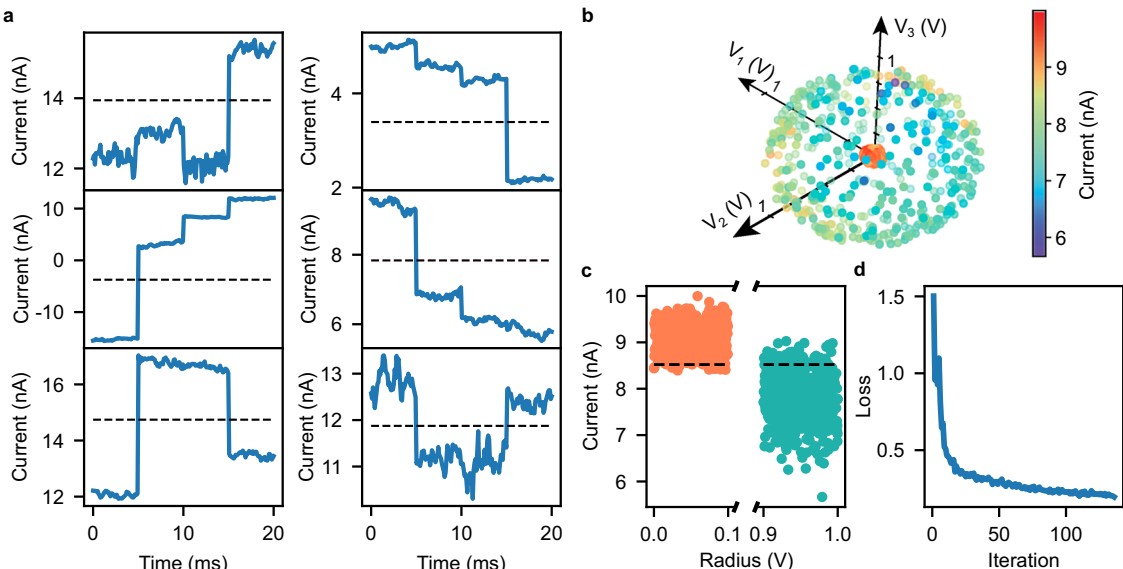

**Fig. 4 | Gradient descent in materia for realizing benchmark tasks. a** Output currents of the optimized Boolean logic gates at 77 K. The black dashed line separates the "high" and "low" logic states. The output signal was processed such that the ramping up and down between the different logic configurations (which takes 15 ms, including waiting time) is removed. **b** 3D scatter plot showing the position of the test data points for the sphere classification in the 3D feature space composed of the voltages of the 3 input electrodes. The colour map indicates the optimized current value corresponding to each point, after the HGE-based training. **c** Scatter plot of the output current values as a function of distance to the sphere centre (radius) for the test data points, after the training. The black dashed line represents the threshold between the 0 and 1 current levels of the two classes (orange and green). **d** Value of the loss function versus the number of iterations of the gradient descent algorithm.

Subsequently, a gradient descent optimization scheme can be used to optimize the control parameters in materia. We demonstrated how the approach can be utilized to obtain Boolean functionality and a sphere classification task in a RNPU. In the present work, we still implemented the gradient extraction on an external digital computer. However, since the implementation only requires simple arithmetic operations and a small memory allocation per input terminal, it is straightforward to realize embedded solutions of our approach without need for external computing. We foresee that our approach will facilitate material-based edge learning, i.e., material-based systems for artificial intelligence that are independent of a centralized training procedure and that can continuously process data in a changing environment. Thus, applications of our method are envisioned for online machine-learning scenarios in edge computing, autonomous adaptive control, and deep reinforcement learning systems where sensory information is directly processed by material-based computing.

## Methods
### Samples
300 nm of thermal oxide was grown on an n-type silicon substrate (resistivity 1–20 Ω cm), in which $26 \times 60\ \mu m^2$ implantation windows were defined by photolithography and wet etching. Another 35 nm of oxide was thermally grown in the implantation window to serve as a stopping layer. After boron implantation (9 keV equivalent, $3.5 \times 10^{14}\ cm^{-2}$), and activation via rapid thermal annealing (1050 °C, 7 s, the 35-nm stopping layer was removed by wet etching. The boron concentration near the silicon surface exceeds $2 \times 10^{19}\ cm^{-3}$ to ensure Ohmic contact with the electrodes and decreases monotonically with depth. Eight 1.5 nm Ti/25 nm Pd nanoelectrodes, with the respective bonding pads, were patterned on top of the silicon by electron-beam lithography and lift-off. The devices were annealed at 160 °C for 10 min to promote the metal/silicon contact quality. The silicon surface was further etched by reactive ion etching to reduce the boron concentration in the active gap area. The surface was finally treated with

mild oxygen plasma, followed by 1% HF etching to remove possible contaminants.

### Measurement setup
During all measurements, the devices were inserted into a liquid-nitrogen (77 K) dewar with a customized dipstick. The input voltages are supplied by a National Instruments cDAQ 9264. The IV converter for the output current was placed on the printed circuit board (PCB) of the device to avoid capacitive crosstalk in the wiring between the dipped device and the output sampler. The output was sampled by a National Instruments cDAQ 9202, with a maximum sample rate of 10,000 Hz. Therefore, the update rate of the cDAQ 9264 was also set to 10,000 Hz.

### Update scheme
An iterative scheme is used to update the control voltages to be optimized, given by

$$\mathbf{V}^{(n+1)} = \mathbf{V}^{(n)} - \eta \nabla E\left(\mathbf{V}^{(n)}\right) \qquad (1)$$

where $\mathbf{V}$ is the vector of control voltages, $E$ is the loss function (see "Methods": "Loss function") and $\eta$ is the learning rate, which is set to 0.02 for the logic gates, and to 0.015 for the sphere classifier. In both cases, the optimization ran for 300 iterations or was halted earlier when $E$ reached a threshold value, which was set to 0.008 for the former tasks, and 0.2 for the latter. The gradient with respect to the control voltages is split into two parts using the chain rule:

$$\nabla E(\mathbf{V}) = \frac{dE}{dI_{out}} \nabla I_{out}(\mathbf{V}) \qquad (2)$$

where $I_{out}$ is the output current, which we will from now on simply call $I$. The first term is straightforward to calculate since we have an analytical expression of the loss function. The second term is determined using HGE. See "Methods": HGE for details.

## Principle of homodyne gradient extraction for a nanoelectronic device

Our HGE approach extracts the derivatives (gradient components) with respect to each input voltage either sequentially or simultaneously by perturbing each input voltage with a sinusoidal perturbation of a distinct frequency. Our nanoelectronic device can be described by a function $I$ which returns the output current for the $N$ input voltages: $I(V_1, V_2, \ldots, V_N)$. We perturb each input voltage sinusoidally with a distinct frequency $f_n$, phase $\phi_n$ and, for generality, a distinct amplitude $\alpha_n$. We assume that the perturbations are small enough that a first-order Taylor expansion of $I$ in the input voltages is valid. The resulting outcome is:

$$
\begin{aligned}
&I(V_1 + \alpha_1 \sin(2\pi f_1 t + \phi_1), V_2 + \alpha_2 \sin(2\pi f_2 t + \phi_2), \ldots, V_N \\
&\quad + \alpha_N \sin(2\pi f_N t + \phi_N)) \\
&\approx I(V_1, V_2, \ldots, V_N) + \frac{\partial I}{\partial V_1}\alpha_1 \sin(2\pi f_1 t + \phi_1) + \frac{\partial I}{\partial V_2}\alpha_2 \sin(2\pi f_2 t + \phi_2) \\
&\quad + \ldots + \frac{\partial I}{\partial V_N}\alpha_N \sin(2\pi f_N t + \phi_N) \\
&\equiv I_0 + I_1 \sin(2\pi f_1 t + \phi_1) + I_2 \sin(2\pi f_2 t + \phi_2) + \ldots + I_N \sin(2\pi f_N t + \phi_N) \\
&= I_0 + \sum_{n=1}^{N} I_n \sin(2\pi f_n t + \phi_n)
\end{aligned}
$$

$$(3)$$

where $I_n$ is the amplitude of the sinusoidal modulation of the output current resulting from perturbing voltage $n$.

As clear from the expression above, the amplitude of the $n$th sinusoidal modulation of the output current is proportional to the $n$th partial derivative, and hence the $n$th component of the gradient $\nabla I(\mathbf{V})$. By making use of the orthogonality of sine waves, we can extract the gradient information of each component separately. This homodyne procedure is similar to what is done in lock-in amplifiers. The measured output signal is multiplied with a reference signal (currently done in software). This reference signal consists of a sine wave with frequency $f_m$ of the perturbation on the electrode $m$ for which we want to determine the derivative. This signal mixing results in the sum and difference frequencies:

$$
\begin{aligned}
&\sum_{n=1}^{N} I_n \sin(2\pi f_n t + \phi_n) \sin(2\pi f_m t + \phi_m) \\
&= \sum_{n=1}^{N} \frac{I_n}{2}\left[\cos(2\pi(f_m - f_n)t + \phi_n - \phi_m) - \cos(2\pi(f_m + f_n)t + \phi_n + \phi_m)\right]
\end{aligned}
$$

$$(4)$$

For $n = m$ the output perturbation corresponding to input electrode $m$ is thus split into a DC component (difference frequency) and the so-called $2f$ component: a sinusoidal function with twice the original frequency $f_m$ (sum frequency). All other components $n \neq m$ consist sinusoidal functions with nonzero frequency. Thus, all contributions to the output current that originate from other input electrodes can straightforwardly be filtered using a low-pass filter (we choose to use a third-order Butterworth filter, implemented in a digital computer, providing a good compromise between speed and accuracy). The only term that remains is that resulting from electrode $m$:

$$
I_{\text{filtered}} \approx \frac{I_m}{2}\cos(\phi_m)
$$

$$(5)$$

If any phase delays can occur in the device, this expression will depend on the phase $\phi_m$. To extract this phase, the output signal is multiplied with a sine wave with a frequency $f_m$, but with a phase-shifted over 90°. As a result, two signals are obtained: an in-phase signal $X = \frac{I_m}{2}\cos(\phi_m)$ and a quadrature signal $Y = \frac{I_m}{2}\sin(\phi_m)$. The

amplitude is then determined as $I_m = \sqrt{X^2 + Y^2}$ and the phase as $\phi_m = \arctan\left(\frac{Y}{X}\right)$ (note that this now represents the output response divided by a factor of 2). If the device or measurement equipment does not cause phase delays, mixing the output with an in-phase reference signal is sufficient to obtain the derivatives. In this case, a single frequency can be applied to two different input electrodes, given that their phase difference is 90°. Their respective reference signal will attenuate the other, phase-shifted, signal when extracting the derivative.

Since the perturbation applied to each input terminal has a distinct frequency and phase combination, we can calculate the values $I_m$ for all terminals $m$ simultaneously from a single measurement of the output signal by repeating the above procedure for each frequency $f_m$. The final step in obtaining the derivatives with respect to each input voltage is dividing the values $I_m$ by (half of) the amplitudes of the perturbations $\alpha_m/2$. The derivatives are then given by $\frac{\partial I}{\partial V_m} = \rho_m \frac{I_m}{\alpha_m}$, where $\rho_m$ is the sign of the gradient, with values 1 or −1 for $\phi_m \in [-90, 90]$ or $\phi_m \in [90, 270]$ degrees, respectively.

To ensure an accurate extraction of the gradient, the following issues must be taken into consideration. First, undesired contributions to $I_{\text{filtered}}$ might have frequencies close to $f_m$. It is therefore crucial to use a low-pass filter with a sufficiently low cut-off frequency. See the next section for a description of the relation between the parameters that determine signal, filter, and noise strength and how these determine the parallelization of the procedure. Second, the frequencies $f_n$ should ideally be chosen in a low-noise frequency domain because otherwise the above procedure will pick up spurious noise contributions. Due to the presence of 1/f-like noise[34] higher perturbation frequencies will in principle, yield more accurate results. On the other hand, the perturbation frequencies should be sufficiently low to be in the static response regime of the device. Finally, the magnitude $\alpha_n$ of the perturbations should be low enough to be in the linear response regime of the device. For too large perturbations, Eq. (3) will no longer hold for the response of the device.

## Analytical description of HGE

Given the assumptions that the output response is linear for small input perturbations and that the dominant noise type around the operating frequencies is Gaussian white noise, HGE can be described analytically by computing the contributions of the individual perturbations to the total output signal. The purpose of the analytical description is to determine how HGE can be efficiently parallelized without introducing additional errors to the extracted single gradient components. The analytical description will furthermore help us understand how HGE compares to the FD method.

The gradient extraction process (i.e., filtering a mixed signal and dividing by a constant) can be described as a linear and time-invariant system. Consequently, HGE can be described by its (power) transfer function, allowing for a straightforward decomposition of the mixed signal into the various components. The power transfer function is used here, since this can directly be used to compute the expected squared error of the gradient extraction procedure. The power transfer function of the mixed signal is given by:

$$
S_Y(f) = |H(f)|^2 S_X(f)
$$

$$(6)$$

where $S_Y(f)$ is the output power spectrum of the filtered signal, $H(f)$ is the frequency response function of the low-pass filter, and $S_X(f)$ is the power spectrum of the device signal, after mixing with the reference signal. For the case of sequential HGE (i.e., applying a perturbation to only a single parameter), the frequencies of $S_Y(f)$ can be split in three parts: the perturbation amplitude that we are trying to extract (shifted to $f = 0$ Hz), the $2f$-component (at twice the reference frequency), and

noise (all remaining other frequencies):

$$S_Y(f) = |H(0)|^2 S_X(0) + |H(2f_m)|^2 S_X(2f_m) + |H(f)|^2 S_{noise}(f)$$
$$= P_m + P_{2f} + P_{noise} = \frac{I_m^2}{4} + |H(2f_m)|^2 \frac{I_m^2}{4} + \frac{f_{enbw}S_N}{2} \quad (7)$$

The power at the perturbed frequency corresponding to perturbing input $m$ is given by $P_m = I_m^2/4$ (see Eq. (3) of the previous section), where $I_m$ is the amplitude at the output of the device. Here, we assume for simplicity that the gain of the low-pass filter is 1, i.e., $|H(0)|^2 = 1$. The power of the $2f$-component is denoted by $P_{2f}$ and the power of the noise is denoted by $P_{noise}$, which can easily be found by making use of the equivalent noise bandwidth (ENBW) frequency, $f_{enwb}$. The ENBW is defined as the bandwidth of a brick-wall filter that produces the same noise power as the actual filter, multiplied by the flat noise power $S_N$. The expected mean squared error in the extracted derivative for input $m$, which is used to compare with the measured derivative in Fig. 2d, is given by:

$$MSE_m = 4\frac{P_{2f} + P_{noise}}{\alpha_m^2} \quad (8)$$

where the undesired signals of Eq. (7) are divided by the square of half the amplitude of the input perturbation $(\alpha_m/2)^2$ to obtain the error in the derivative.

When HGE is used in parallel, an additional term appears in the power spectrum in Eq. (7), which is the contribution of the parallel frequencies, $P_p$. Since the parallel frequencies are pre-defined by the user, the shifted frequencies caused by the signal mixing of the output signal with the reference signal are known, and therefore we precisely know much they are attenuated by the low-pass filter. The power signal is given by

$$P_p = \sum_{n=1}^{N} \frac{I_n^2}{4} \left( |H(f_m + f_n)|^2 + |H(f_m - f_n)|^2 \right) \quad (9)$$

where $I_n^2/4$ is the unfiltered power of the output signal at parallel frequency $f_n$, and $H(f_m \pm f_n)$ is the corresponding frequency response of the lowpass filter at $f_m \pm f_n$. Analogous to the error terms in the sequential case, to determine the biasing impact of the parallel frequencies on the derivative estimation for input signal $m$, Eq. (9) should be divided by $(\alpha_m/2)^2$. The corresponding expected error for the parallel case is thereby given by:

$$MSE_{p,m} = 4\frac{P_p + P_{2f} + P_{noise}}{\alpha_m^2} \quad (10)$$

To use Eqs. (8) and (10) in practice, we need to attribute values to the parameters ($S_N$, $I_m$, and $I_n$). This can be straightforwardly done by performing simple measurements, which need to be done only once. The flat noise power, $S_N$, can be obtained by applying a steady input signal to the device for a few seconds and transforming the output signal into a power spectrum. The typical values for the output responses, $I_m$ and $I_n$, can be estimated by perturbing the input parameters while sampling points throughout the input space. These are mainly necessary for the parallel case, in which they are used to determine an upper limit for the estimated MSE (see Fig. 3b). See Supplementary Note 2 for the results of sampling the input space.

### Loss function

The loss function to obtain Boolean logic functionality (the same as used in ref. 12) is given by $E(y, z) = (1 - \rho(y, z))/\sigma(y_{sep})$, where $y$ are the measured currents, $z$ the targeted currents, $\rho(y, z)$ is the Pearson correlation coefficient and $\sigma$ the sigmoid function. The value $y_{sep}$ represents the average separation between the high and low labelled data.

While the correlation function promotes similarity between the targeted and predicted outputs, the sigmoid function promotes separation between the two logic current levels.

## Data availability

The data presented here are available at https://doi.org/10.5281/zenodo.17136119.

## Code availability

The custom computer code used here is available under the GNU General Public License v3.0 at https://github.com/BraiNEdarwin/brains-py. The code for recreating the results can be found at https://github.com/Mark-Boon/code_GD_in_materia_Boon25.

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

## Acknowledgements

We thank B.J. Geurts and H.J. Broersma for fruitful discussions. We thank M.H. Siekman and J.G.M. Sanderink for technical support. We acknowledge financial support from the University of Twente (W.G.v.d.W.), the Deutsche Forschungsgemeinschaft (DFG, German Research Foundation) under Germany's Excellence Strategy—EXC 2002/1 "Science of Intelligence"—project number 390523135 (M.N.B.) and SFB 1459/2 2025—433682494 (L.C. and W.G.v.d.W.), the Dutch Research Council (Natuurkunde Projectruimte grant no. 680-91-114 and HTSM grant no. 16237) (U.A.I) and Toyota Motor Europe N.V. (U.A.I.).

## Author contributions

M.N.B. performed the simulations, L.C. performed the measurements with the help of M.N.B. M.N.B. designed the experiments. B.v.d.V. and T.C. fabricated the samples. U.A.I. helped to develop the general software framework. M.N.B., L.C., H.-C.R.E., and W.G.v.d.W. wrote the manuscript. T.C., H.-C.R.E., and W.G.v.d.W. conceived the project. W.G.v.d.W. and P.A.B. supervised the project.

## Funding

## Competing interests

The authors declare no competing interests.
