## [Transparent Peer Review file · Nature Communications]

Gradient descent *in materia* through homodyne gradient extraction

Corresponding Author: Professor Wilfred van der Wiel

Version 0:

Reviewer comments:

Reviewer #1

(Remarks to the Author)

The authors have not made the review easy for us, as the response document does not clearly reference the specific manuscript changes, and neither the revised manuscript nor the supplementary note is marked to highlight these revisions. Given the very substantial time since the original review, identifying the new content required a bit of effort.

Initially, I recommended rejection despite appreciating the paper, because I felt the method lacked broad applicability, originality was somewhat limited compared to existing techniques, and the experimental realization was impressive but not necessarily a masterpiece. I still feel the paper is not a Nature paper, but for Nature Communications, it is certainly suitable.

The new additions, although relatively minimal, do provide useful clarity. I would have preferred if Supplementary Note 4 included experimental or simulation results. Nevertheless, I recommend publication in its current form.

Reviewer #4

(Remarks to the Author)

The authors present a gradient estimation method for black-box systems that enables learning by directly minimizing a loss function. The core technique, referred to as “homodyne gradient extraction (HGE),” relies on applying sinusoidal perturbations to the input and analyzing how these perturbations influence the loss. By observing the system’s response, these variations reveal how the system should adapt its input to move toward lower loss values. This method is experimentally validated by training a dopant-based material network to perform logical operations and sphere classification.

Although the core of the proposed approach (HGE) is conceptually similar to the Simultaneous Perturbation Stochastic Approximation (SPSA) method, the lower variance of gradient estimates obtained via HGE—especially when compared to finite-difference (FD) methods in noisy physical systems—is particularly interesting. The authors applied their method to relatively simple AI tasks, which is acceptable, provided they demonstrate that the estimated gradients are reasonably aligned with the true gradients and include comparisons to SPSA. Therefore, the reviewer recommends that the authors place greater emphasis on evaluating the accuracy of the estimated gradient vectors—using metrics such as relative error and cosine similarity—and directly compare their results with those obtained using SPSA before publication.

- As mentioned above, in the case of parallelized HGE—which is one of its main advantages—the comparison should be made against SPSA rather than finite-difference (FD) methods, particularly in terms of gradient variance and computational speedup. SPSA perturbs all input coordinates simultaneously using a random $\pm\delta$ vector and computes the gradient using a symmetric difference. Notably, it requires only two function evaluations per iteration, regardless of the input dimensionality, and is known for its robustness to noise.

It would therefore be highly valuable to include a direct comparison with SPSA in the multiple-input scenario. For instance, if the authors demonstrate a 7× speedup using 7 input channels in the DNPU device compared to the FD method, a fair benchmark would be to compare against SPSA, rather than FD.

The authors could assess whether their method offers clear advantages over SPSA in terms of speedup, query budget (i.e., the number of forward evaluations), and gradient estimation performance.

- Measurement time is also an important factor for HGE. As shown in Fig. 2d, the variance of the gradients estimated using HGE decreases as the measurement time increases. This suggests that relatively long measurement durations are required for HGE to achieve low-variance gradient estimates—unlike the FD method, which does not exhibit the same dependency on measurement time. A natural question arises: under the same total measurement time (e.g., $T = 0.1$ seconds), how does the gradient variance change if the number of forward evaluations is increased only for the FD method? More specifically, if we fix the total measurement time for both FD and HGE, but increase the number of forward evaluations only for FD, it would be useful to examine how this impacts the accuracy and variance of the gradient estimates. This comparison could help clarify the trade-offs between sampling strategy and time efficiency across methods.

- Another important concern that warrants investigation is the scalability of the proposed method to high-dimensional black-box systems, particularly in terms of the number of inputs and outputs. To assess this, the authors could analyze the norm of the estimated gradients—both the mean and variance—across increasing dimensionalities in a controlled simulation environment.

For instance, a well-defined, high-dimensional system with access to the true gradients—such as a network of coupled nonlinear oscillators—could serve as a useful testbed. By introducing noise into the system and comparing the estimated gradients to the known true gradients, the authors could quantitatively evaluate how well their method scales and maintains accuracy in more complex settings.

- The sentence in line 89 does not appear to be fully correct to the reviewer: “Therefore, it could be challenging to apply PLL to disordered systems and devices that have recently become increasingly popular [15][16], due to the necessity of further design considerations.”

The authors are encouraged to revise this sentence for clarity and completeness. The authors could mention that Physical Local Learning (PLL) is an approach for training deep physical neural networks composed of multiple physical layers, each optimized using local loss functions. However, updating the parameters of each physical layer requires accurate estimation of gradient vectors, which can be particularly challenging in disordered systems.

Version 1:

Reviewer comments:

Reviewer #4

(Remarks to the Author)

The response of the authors is satisfactory.

REVIEWER COMMENTS

Reviewer #1 (Remarks to the Author):

The authors have not made the review easy for us, as the response document does not clearly reference the specific manuscript changes, and neither the revised manuscript nor the supplementary note is marked to highlight these revisions. Given the very substantial time since the original review, identifying the new content required a bit of effort.

Initially, I recommended rejection despite appreciating the paper, because I felt the method lacked broad applicability, originality was somewhat limited compared to existing techniques, and the experimental realization was impressive but not necessarily a masterpiece. I still feel the paper is not a Nature paper, but for Nature Communications, it is certainly suitable.

The new additions, although relatively minimal, do provide useful clarity. I would have preferred if Supplementary Note 4 included experimental or simulation results. Nevertheless, I recommend publication in its current form.

We sincerely apologize for not making the review easier. However, we are delighted that Reviewer #1 recommends publication in its current form.

Reviewer #4 (Remarks to the Author):

The authors present a gradient estimation method for black-box systems that enables learning by directly minimizing a loss function. The core technique, referred to as “homodyne gradient extraction (HGE),” relies on applying sinusoidal perturbations to the input and analyzing how these perturbations influence the loss. By observing the system’s response, these variations reveal how the system should adapt its input to move toward lower loss values. This method is experimentally validated by training a dopant-based material network to perform logical operations and sphere classification.

Although the core of the proposed approach (HGE) is conceptually similar to the Simultaneous Perturbation Stochastic Approximation (SPSA) method, the lower variance of gradient estimates obtained via HGE—especially when compared to finite-difference (FD) methods in noisy physical systems—is particularly interesting. The authors applied their method to relatively simple AI tasks, which is acceptable, provided they demonstrate that the estimated gradients are reasonably aligned with the true gradients and include comparisons to SPSA. Therefore, the reviewer recommends that the authors place greater emphasis on evaluating the accuracy of the estimated gradient vectors—using metrics such as relative error and cosine similarity—and directly compare their results with those obtained using SPSA before publication.

We thank Reviewer #4 for the recommendation. We agree that a detailed comparison between HGE and SPSA is essential to validate our approach. As suggested, we have added a simulation-based evaluation of the estimated gradients in a system of coupled oscillators. The quality of single-gradient estimates is assessed using relative error as a metric, with both methods benchmarked on this system. In our analysis, we directly compare HGE with multiple SPSA variants under varying conditions, including the dimensionality of the benchmark task

and the power of the additive noise. To ensure a convincing comparison between the methods, we included three SPSA variants: one using 2 samples to estimate the gradient; a second using 10 samples to estimate 5 independent gradients (*i.e.*, in 5 random directions), which were then averaged; and a third using the same number of samples as HGE to estimate the average gradient. This last variant matches HGE in per-iteration computational cost.

We find that across a broad range of conditions, varying the dimensionality from 5 to 500 parameters and the noise power from 1 Hz^{-1} to 100 Hz^{-1} , HGE consistently outperforms SPSA. In terms of single-gradient estimation accuracy, HGE clearly outperforms all SPSA variants, including the one matched in computational cost (see Supplementary Figure 4). Only when the number of optimizable parameters is very low do the 2- and 10-sample SPSA variants potentially outperform HGE. More importantly, HGE normally requires fewer overall samples to reach convergence on the benchmark task (see Supplementary Figure 5). Again, only in very low-dimensional tasks do the 2- and 10-sample SPSA variants show a possible advantage. However, these SPSA methods do not scale well with dimensionality and fail to find suitable solutions in higher-dimensional spaces.

We would like to highlight key differences between HGE and SPSA that are likely to contribute to HGE's superior performance. While both are zeroth-order (ZO) optimization methods, they estimate gradients in fundamentally different ways, resulting in distinct properties. Unlike SPSA, HGE perturbs each input dimension explicitly and offers full control over optimization hyperparameters. This includes the ability to set individual step sizes per input and to modulate the gradient in the frequency domain. Specifically, HGE allows us to assign distinct frequency components to specific parameters and to control the density of frequency allocation. This flexibility is particularly valuable when dealing with non-flat noise spectra (*e.g.*, $1/f$ noise) or when certain frequency bands are unusable due to external interference, a common issue in hardware implementations.

Additionally, in systems where some parameters are more sensitive and require more accurate gradient estimation, HGE enables tailored tuning of hyperparameters to accommodate these differences, making it well-suited for adaptive optimization.

- As mentioned above, in the case of parallelized HGE—which is one of its main advantages—the comparison should be made against SPSA rather than finite-difference (FD) methods, particularly in terms of gradient variance and computational speedup. SPSA perturbs all input coordinates simultaneously using a random $\pm\delta$ vector and computes the gradient using a symmetric difference. Notably, it requires only two function evaluations per iteration, regardless of the input dimensionality, and is known for its robustness to noise. It would therefore be highly valuable to include a direct comparison with SPSA in the multiple-input scenario. For instance, if the authors demonstrate a $7\times$ speedup using 7 input channels in the DNPU device compared to the FD method, a fair benchmark would be to compare against SPSA, rather than FD.

The authors could assess whether their method offers clear advantages over SPSA in terms of speedup, query budget (*i.e.*, the number of forward evaluations), and gradient estimation performance.

We thank Reviewer #4 for emphasizing the importance of comparing HGE and SPSA in parallelized settings. As detailed above and in the revised Supplementary Information, we evaluated multiple SPSA variants (2-sample, 10-sample, and N -sample) against adaptive N -sample HGE. HGE consistently showed lower relative error across dimensionalities and noise levels, supporting its advantage in noisy, parallelizable environments.

- Measurement time is also an important factor for HGE. As shown in Fig. 2d, the variance of the gradients estimated using HGE decreases as the measurement time increases. This suggests that relatively long measurement durations are required for HGE to achieve low-

variance gradient estimates—unlike the FD method, which does not exhibit the same dependency on measurement time. A natural question arises: under the same total measurement time (e.g., $T = 0.1$ seconds), how does the gradient variance change if the number of forward evaluations is increased only for the FD method? If HGE, but increase the number of forward evaluations only for FD, it would be useful to examine how this impacts the accuracy and variance of the gradient estimates. This comparison could help clarify the trade-offs between sampling strategy and time efficiency across methods.

We agree that comparing measurement time and gradient estimation quality is essential for assessing efficiency under fixed time budgets. As noted above, this is closely related to the comparison in terms of speedup, sample budget, and gradient estimation performance. To address this, we focused on SPSA rather than FD, as SPSA is better suited for zeroth-order optimization in high-dimensional and noisy scenarios. As mentioned above, we benchmarked multiple SPSA variants against HGE under controlled conditions, demonstrating that HGE consistently achieves higher gradient accuracy within comparable or lower measurement time.

- Another important concern that warrants investigation is the scalability of the proposed method to high-dimensional black-box systems, particularly in terms of the number of inputs and outputs. To assess this, the authors could analyze the norm of the estimated gradients—both the mean and variance—across increasing dimensionalities in a controlled simulation environment.

For instance, a well-defined, high-dimensional system with access to the true gradients—such as a network of coupled nonlinear oscillators—could serve as a useful testbed. By introducing noise into the system and comparing the estimated gradients to the known true gradients, the authors could quantitatively evaluate how well their method scales and maintains accuracy in more complex settings.

See our replies above.

- The sentence in line 89 does not appear to be fully correct to the reviewer: “Therefore, it could be challenging to apply PLL to disordered systems and devices that have recently become increasingly popular [15][16], due to the necessity of further design considerations.” The authors are encouraged to revise this sentence for clarity and completeness. The authors could mention that Physical Local Learning (PLL) is an approach for training deep physical neural networks composed of multiple physical layers, each optimized using local loss functions. However, updating the parameters of each physical layer requires accurate estimation of gradient vectors, which can be particularly challenging in disordered systems.

We agree that the sentence, in its original formulation, does not provide a sufficiently clear explanation to why PLL is not well-suited for optimizing disorder systems. Thus, we apologize for the inaccuracy, and we are grateful to Reviewer #4 for pointing that out. We changed the text based on the Reviewer's advice.